# What motivates people with type 2 diabetes to maintain lifestyle changes and what challenges do they experience? A qualitative evidence synthesis

Tina Bekken◉ , Nina Ottesen◉*, Claire Glenton‡, Lena Victoria Nordheim‡

Department of Health and Functioning, Western Norway University of Applied Sciences, Bergen, Norway

◉ These authors contributed equally to this work.
‡ CG and LVN also contributed equally to this work.
* tinabekken@hotmail.com

## Abstract

### Background

People with type 2 diabetes often find recommended lifestyle changes difficult to achieve and can find it particularly difficult to maintain these changes. It is therefore useful to explore their views and experiences about what influences them when attempting to achieve these goals.

### Objectives

To identify, appraise and synthesise qualitative studies that explore what motivates people with type 2 diabetes to maintain lifestyle changes over time and the challenges they experience.

### Methods

We included qualitative studies and mixed methods studies with a qualitative component, in any language, of adults who had type 2 diabetes and who had maintained lifestyle changes for a minimum of twelve months.
We searched MEDLINE, CINAHL, PsycINFO and EMBASE from inception to 4ᵗʰ November 2024. We extracted data using a pre-designed form and assessed methodological limitations using predefined criteria. We used a thematic synthesis approach to analyse the data. We assessed our confidence in the review findings using the GRADE-CERQual approach. Finally, we used our findings to develop a model using the locus of control theory and to develop implications for practice.

**Data availability statement:** All relevant data are within the manuscript and its Supporting Information files.

**Funding:** The author(s) received no specific funding for this work.

**Competing interests:** The authors have declared that no competing interests exist.

## Results

We included twelve studies from Europe, USA, Canada, South-Africa, Australia and New Zealand. Participants described several factors that motivated them to maintain lifestyle changes, including getting support from others, seeing results from their lifestyle changes, walking as an activity, being physically active with others, fearing the consequences of type 2 diabetes, and taking control. Factors that they viewed as challenging included mental, physical, practical, and social challenges around exercise, and lack of follow-up from healthcare professionals.

## Conclusions

Our review identified several factors that may influence maintenance of lifestyle changes in type 2 diabetes. Based on these findings, we have developed a series of questions to help people with type 2 diabetes and their healthcare providers think about how best to maintain lifestyle choices over time.

## Introduction

Type 2 diabetes is a chronic, progressive disease with an increasing impact on individuals, populations, and public health. Type 2 diabetes affects more than 425 million people worldwide and accounts for over 90% of all diabetes diagnoses [1].

Type 2 diabetes is associated with increased mortality and morbidity [2]. It is possible to slow and reverse disease progression through lifestyle changes [3–6] and this is seen as the main strategy for overcoming the type 2 diabetes epidemic [7]. However, research indicates that lifestyle changes are difficult to achieve and maintain [8] and finding ways to support adherence has been identified as an important challenge for the future [9]. It is therefore useful to explore the views and experiences of people with type 2 diabetes about what motivates and challenge them to maintain lifestyle changes.

The locus of control theory [10] has been used in several studies when assessing people's adherence to diabetes regimens [11–12]. Here, researchers explore how different people perceive their ability to control their lives, the extent to which they believe it is controlled by themselves ('internal control') or by outside forces ('external control') and how this influences their personal health. Some studies argue that people with diabetes can stick more tightly to a regime if they experience an increase in internal control [12]. According to Rodin (1986) [13], a person with a high perception of control may have better health because he or she is likely to make decisions that promote health.

Several trials and systematic reviews have assessed the effect of various interventions to change and maintain lifestyle among people with type 2 diabetes [14–16]. In addition, one qualitative evidence synthesis has investigated the maintenance of lifestyle change in people with prediabetes [17] the everyday challenges faced by people with type 2 diabetes [18], and how they see self-help strategies as helpful or not [19], respectively. However, to our knowledge, there are no qualitative evidence

syntheses focusing on factors that contribute to the long-term maintenance of lifestyle changes. Our main objective is to synthesize existing qualitative studies to explore what motivates people with type 2 diabetes to maintain lifestyle changes over time and the challenges they experience.

## Methods

The protocol for this qualitative evidence synthesis was registered in PROSPERO id: CRD42023395980.

When preparing this review, we used the Cochrane Effective Practice and Organisation of Care template for qualitative evidence synthesis [20], the ENTREQ statement (Enhancing Transparency in Reporting the synthesis of Qualitative research) [21] (see S1 Appendix: ENTREQ Checklist) and the PRISMA 2020 statement (Preferred Reporting Items for Systematic reviews and Meta-Analyses) [22] (see S2 Appendix: PRISMA 2020 Checklist).

### Criteria for considering studies for this review

**Topic of interest.** We included studies where the main topic of interest was the views and experiences of people with type 2 diabetes in maintaining lifestyle changes over time. For the purposes of this review, we defined key concepts as follows:

- Lifestyle change: changed lifestyle habits, such as changing diet, increased physical activity, and smoking cessation.

- Motivational factors: self-perceived factors that initiate and maintain a change in behaviour over time.

- Maintaining lifestyle change: changes that are kept up for 12 months or more.

**Types of participants.** We included studies of participants over 18 years of age with type 2 diabetes. We excluded studies containing data from people with other forms of diabetes.

**Types of studies.** We included primary studies in any language that use qualitative study designs such as ethnography, phenomenology, grounded theory, case studies, and qualitative process evaluations. We included studies that use both qualitative methods for data collection and qualitative methods for data analysis. We included mixed methods studies where it was possible to extract the data that were collected and analysed using qualitative methods.

### Search methods for identification of studies

We searched four electronic databases from their inception through November 4,2024: MEDLINE, PsycINFO, EMBASE (all via Ovid) and CINAHL (EBSCO). We developed a search strategy for MEDLINE, using subject headings and text words related to type 2 diabetes and lifestyle change, including concepts such as lifestyle, health behaviour, adherence to changes, weight reduction, diet, physical activity and smoking cessation. The search strategy was modified for the other databases. We used validated filters for identifying qualitative studies in each database [23–25]. We did not apply any date or language limits. See S3 Appendix for the complete search strategy.

We conducted citation searches in Web of Science (completed February 19, 2025) and searched the reference lists of included studies. We did not search for grey literature or unpublished studies.

### Selection of studies

Two pairs of review authors (TB and NO, or CG and LVN) independently assessed the titles and abstracts of the identified records to evaluate eligibility. We retrieved full text of all papers identified as potentially relevant by either pair of reviewers. Both reviewers then assessed these papers independently. Disagreements were resolved by discussion or, when required, by involving a third person (LVN or TB/NO). We used the DistillerSR software in the selection process [26]. We used Google Translate for abstracts written in other languages than English or the Scandinavian languages. We had a predefined plan to get studies translated into English where it was not possible to read the abstract or full-text in English.

## Data extraction

Two review authors (TB, NO) extracted information about the included studies, including the objectives of the study, information about participants, setting, context, method, results, and author's conclusion. The study findings were extracted by TB and NO independently. Any discrepancies were discussed until we reached agreement or by involving supervisors (CG and LVN). We did not obtain supporting or missing information by contacting the authors of the original study reports.

## Assessing the methodological limitations of included studies

Two review authors (TB, NO) independently assessed methodological limitations for each study using a list of criteria that has been used in previous qualitative evidence syntheses [27,28]. This list of criteria was originally based on the Critical Appraisal Skills Programme (CASP) tool [29] but has since gone through several iterations. We assessed methodological limitations according to the criteria specified in Table 2. Disagreements were resolved by discussion or, when required, by involving a third person (LVN).

## Data management, analysis and synthesis

We compiled all relevant data from the studies, such as participant quotations or study authors' summaries and descriptions of participants' accounts, into an Excel spreadsheet (see S4 Appendix: Findings supporting data from studies). We initially considered performing a synthesis based on meta-aggregation [30] but since the findings across studies were descriptive, we chose the thematic synthesis approach by Thomas and Harden [31]. This approach is comprised of three phases. In phase 1 we coded the result section of each study line-by-line. In phase 2, through consensus, we looked for similarities between the codes and structured them into descriptive themes according to their meaning and content. In phase 3, we organized the descriptive themes into analytical themes. We did this in two ways. First, we divided the descriptive themes into two categories: 'motivating factors' and 'challenges'. Second, based on the locus of control theory [10,32] we categorised the descriptive themes as factors that people appeared to perceive as a result of their own decisions or actions, or as beyond their control (Fig 2).

Finally, we used the review findings assessed to be of high or moderate confidence to develop implications for practice. We assessed the relevance of these implications by gathering feedback from relevant stakeholders (one general practitioner and one person with type 2 diabetes) and edited our original draft of the implications for practice in response to their feedback.

## Assessing our confidence in the review findings

Two review authors (TB, NO) used the GRADE-CERQual ('Confidence in the Evidence from Reviews of Qualitative research') [33] approach to assess our confidence in each finding based on the following four key components:

1. Methodological limitations of included studies: the extent to which there are concerns about the design or conduct of the primary studies that contributed evidence to an individual review finding.

2. Coherence of the review finding: an assessment of how clear and cogent the fit is between the data from the primary studies and a review finding that synthesises those data. By cogent, we mean 'well supported' or 'compelling'.

3. Adequacy of the data contributing to a review finding: an overall determination of the degree of richness and quantity of data supporting a review finding.

4. Relevance of the included studies to the review question: the extent to which the body of evidence from the primary studies supporting a review finding is applicable to the context (perspective or population, phenomenon of interest, setting) specified in the review question.

After assessing each of the four components, we made a judgement about the overall confidence in the evidence supporting the review finding. We judged confidence as high, moderate, low, or very low. The final assessment was based on consensus among the review authors. All findings started as high confidence and were then downgraded if important concerns regarding any of the GRADE-CERQual [32] components were found.

### Review author reflexivity

As recommended in qualitative research [34], we have reflected on how our backgrounds and perspectives may have influenced the design and performance of this research. TB is an occupational therapist with a master`s degree in evidence based practice and is employed at a specialist hospital in rehabilitation and physical medicine. TB works primarily with rehabilitation traumatic brain injury and stroke but meets people who have type 2 diabetes as an additional diagnosis. NO is a radiographer with a master`s degree in evidence based practice and is employed in the pharmaceutical industry (Astra-Zeneca). NO has no contact with patients but works directly with healthcare professionals as a product manager for glucose lowering drugs. TB and NO received supervision and support from CG and LVN. CG and LVN teach evidence-based practice and systematic review methodology but have no particular knowledge or experience in the topic area.

Both TB and NO have good knowledge of the patient population, the disease, and the complications that the disease can cause. We have both worked with lifestyle changes in diet and exercise and believe that these are important areas to prioritize for people with type 2 diabetes. In our experience, lifestyle changes and especially keeping these changes over time, can be difficult. We also believe that a person's understanding of whether their health is controlled by their own behaviour or forces outside of themselves is important for maintaining lifestyle change.

Through this process of creating a qualitative evidence synthesis we have tried to be open to other perspectives than our own and have sometimes been surprised by some of the study results. For instance, the finding that people's motivations to maintain lifestyle changes was partly driven by fears of the consequences of the disease was unexpected. The findings about lack of information, knowledge, and follow-up from healthcare professionals also engaged us and led to many discussions, as well as a desire for the situation to improve.

## Results

The literature search identified 4101 unique references. Following title and abstract screening, we assessed 128 reports in full-text for eligibility. Of these, we excluded 115 reports that potentially met our inclusion criteria. We provide reasons for exclusion in S5 Appendix. Table of excluded studies. The predominant reason for excluding studies was that they did not focus on maintaining lifestyle changes for at least a year, or it was unclear if they did. We included twelve studies in our review [36,38–48]. One study was reported in two publications: a scientific article [36] and a doctoral thesis [37], respectively. We used the former as the primary source for the review, and consulted the latter when information was lacking or unclear. The studies were published between March 2003 and June 2024, in English. The study selection process is presented in Fig 1 in a PRISMA (34) flow diagram.

### Description of the studies

All twelve studies explored the views and experiences of people with type 2 diabetes regarding their maintenance of lifestyle changes for 12 months or more.

The studies were from: the United States (three studies) [36,38,39], Scotland [40], Denmark (two studies) [41,42], Netherlands [43], Norway [44] Australia [45], New Zealand [46], Canada [47] and South Africa [48]. With the exception of one study from a middle-income country [48], all other studies were from high-income countries [36,38–47]. Ten studies included both men and women aged 20 years to 82 years [36,40–42], while one study included only men [38] and women [39], respectively. The studies comprised in total 267 participants. Table 1 presents characteristics of the included studies. See S6 Appendix for further details of study characteristics.

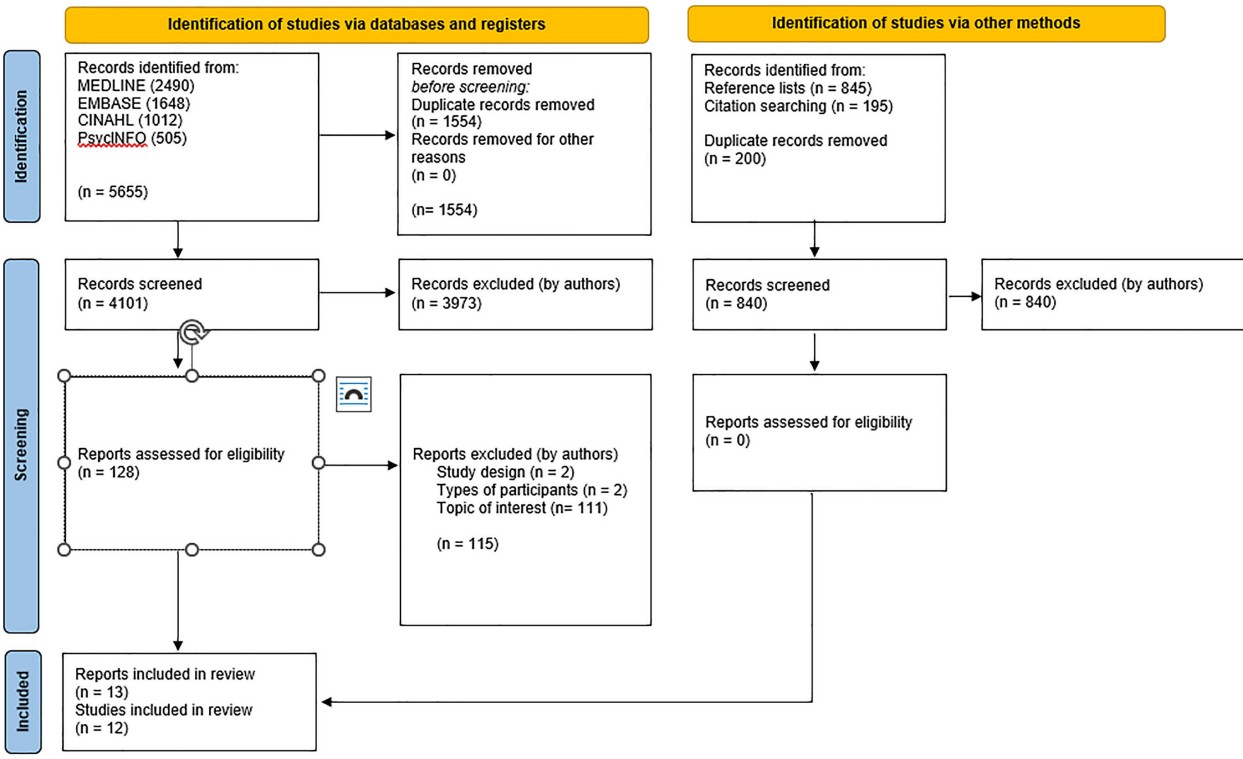

PRISMA 2020 flow diagram for new systematic reviews which included searches of databases, registers and other sources

From: Page MJ, McKenzie JE, Bossuyt PM, Boutron I, Hoffmann TC, Mulrow CD, et al. The PRISMA 2020 statement: an updated guideline for reporting systematic reviews. BMJ 2021 ;372:n71. doi: 10.1136/bmj.n71.

**Fig 1. PRISMA flow diagram.**

## Methodological limitations of the studies and the review

We identified twelve relevant studies for this review. While we searched key databases, it is possible that that searching other databases would have identified additional studies.

We found poor reporting of researcher reflexivity across many of the studies. All studies gave some description, although sometimes very brief, of the context, participants, sampling, methods, and analysis. We assessed nine studies to have minor limitations [38,39,41–44,46–48], and three studies had minor to moderate limitations [36,40,45]. Details of the assessments of methodological limitations for individual studies can be found in Table 2. See S6 Appendix for further elaborations of methodological limitations.

## Confidence in the review findings

Using the GRADE-CERQual [32] approach, we assessed three findings as high confidence, eight findings as moderate confidence and two findings as low confidence.

The main reasons for downgrading our confidence in the findings were concerns related to the relevance of the data (the studies were only from middle and high-income countries in Europe, USA, Canada, Australia, New Zealand, and South Africa), the adequacy of the data (some findings were based on few studies, few participants and some of the findings were supported by data that was thin or only from a small number of studies), and methodological limitations (in particular, concerns about a lack of study author reflexivity). A summary of the overall GRADE-CERQual assessment for each

**Table 1. Characteristics of the included studies.**

| Authors, year, title | Country | Participants | Study design | Method (data collection, analysis) | Research question/purpose |
|---|---|---|---|---|---|
| | | | | (As described by the study authors) | |
| Hall et al., 2003 [36] | New England, USA | 5 in total, 3 men, 2 women | Descriptive qualitative study | In-depth interviews Index case, comparison across cases | "This study describes how individuals with type 2 diabetes overcame obstacles that interfered with maintaining behaviour changes in diet, exercise, and self-monitoring of blood glucose (SMBG)" |
| Peel et al., 2010 [40] | Lothian region, Scotland | 20 in total, 11 men and 9 women | Longitudinal qualitative study | In-depth interviews A critical realist epistemological position. | "To explore type 2 diabetes patients' talk about implementing and sustaining physical activity." |
| Wycherley et al., 2011 [45] | Australia | 30 in total, 22 men and 8 women | Qualitative study | Semi-structured interview | "The purpose of the study was to identify factors that enhance or impede the maintenance of healthy lifestyle behaviors in overweight and obese individuals with type 2 diabetes following a 16-week supervised lifestyle intervention program." |
| Conlin, 2014 [38] | USA | 4 men | Interpretive phenomenological qualitative study | In-depth interviews Interpretative Phenomenological Analysis (IPA). | "What is the lived experience of individuals diagnosed with type 2 diabetes who have made successful lifestyle changes to achieve long-term positive health outcomes?" |
| Phelps, 2014 [39] | USA | 4 women | Interpretive phenomenological qualitative study | In-depth interviews Interpretative Phenomenological Analysis (IPA). | "What is the lived experience of individuals diagnosed with type 2 diabetes who have made successful lifestyle changes to achieve long-term positive health outcomes?" |
| Walker at al., 2018 [41] | Denmark | 5 in total, 4 men and 1 woman | Longitudinal qualitative study | Semi-structured individual Interview Systematic text condensation The framework of self-determination theory was applied to guide analysis | "To explore motivational factors for initiating, implementing, and maintaining physical activity following a rehabilitation program for patients with type 2 diabetes mellitus." |
| Ribu et al., 2019 [44] | Norway | 26 in total | Grounded theory approach with a constant comparative method | Face-to-face, open-ended, in-depth interviews Grounded theory approach, aligning data analysis with ongoing data collection. | "To determine whether the use of a mobile phone-based self-management system for type 2 diabetes for 1 year, with or without telephone health counseling, could improve HbA1c levels, self-management, and health-related quality of life compared to usual care." |
| Schmidt et al., 2020 [42] | Denmark | 6 in total, 3 men and 3 women | Longitudinal qualitative study | In-depth interviews Text condensation with an inductive approach. | "The purpose of this study was to explore and identify factors that influence motivation for and barriers to adopting and maintaining lifestyle changes in patients with type 2 diabetes following participation in an intensive multiple-lifestyle intervention." |
| Jansen et al., 2023 [47] | Canada | 12 in total, 5 men and 7 women | Mixed method | Interviews via telephone at one, six and twelve months post-program. Thematic analysis | "The study aimed to assess changes in glycemic control, blood pressure, anthropometric measures, and physical function after completing an education and exercise program, and to understand the experiences and motivations of individuals with T2D one year after the program." |
| Muchiri et al., 2024 [48] | South Africa | 43 in total, 8 men and 35 women | Phenomenological qualitative study | Focus group discussions and individual interviews. Thematic framework. | "The research aimed to investigate how a randomized controlled trial of an adapted diabetes nutrition education program was received by adults with sub-optimally controlled type 2 diabetes, and to understand the factors influencing the program's outcomes and participant retention." |

*(Continued)*

**Table 1.** (Continued)

| Authors, year, title | Country | Participants | Study design | Method (data collection, analysis) | Research question/purpose |
|---|---|---|---|---|---|
| | | | | (As described by the study authors) | |
| **Van den Burg et al., 2024 [43]** | Nether-lands | 100 in total, 48 men and 44 women | Mixed method | Focus group discussions. Theoretical Domains Framework | *"To investigate whether following a fasting-mimicking diet (FMD) program could influence lifestyle in patients with type 2 diabetes, particularly with regard to diet quality and physical activity."* |
| **Campbell et al., 2024 [46]** | New Zealand | 40 in total, 12 with T2D interviewed at 12 months (4 men and 8 women) | Qualitative study | Semi-structured interview | *"Explore the experiences and acceptability of the Diabetes Remission Clinical Trial (DiRECT) intervention among a predominantly Māori and Pacific Island population living with type 2 diabetes or prediabetes in Aotearoa New Zealand."* |

**Table 2. Assessment of methodological limitations in the included studies.**

| | Setting and context is described sufficiently | Selection strategy is described and appropriate | Data collection strategy is described and justified | Data analysis is described and appropriate | The claims/ findings are supported by sufficient evidence | There is proof of reflexivity | The study shows sensitivity to ethical concerns | There are other concerns | Assessment of methodological limitations |
|---|---|---|---|---|---|---|---|---|---|
| Hall et al., 2003 [36] | Yes | Can`t tell | Yes | Yes | Can`t tell | No | Can't tell | No | Minor to moderate |
| Peel et al., 2010 [40] | Yes | Yes | Yes | Yes | Yes | Can`t tell | Yes | No | Minor to moderate |
| Wycherley et al., 2011 [45] | Can't tell | Yes | Yes | Yes | Can't tell | No | Yes | No | Minor to moderate |
| Conlin 2014 [38] | Yes | Yes | Yes | Yes | Yes | Can`t tell | Yes | No | Minor |
| Phelps 2014 [39] | Yes | Yes | Yes | Yes | Yes | Can`t tell | Yes | No | Minor |
| Walker et al., 2018 [41] | Yes | Yes | Yes | Yes | Yes | Can`t tell | Yes | No | Minor |
| Ribu et al., 2019 [44] | Yes | Yes | Yes | Yes | Yes | No | Yes | No | Minor |
| Schmidt et al., 2020 [42] | Yes | Yes | Yes | Yes | Yes | Can`t tell | Yes | No | Minor |
| Janssen et al., 2023 [47] | Can't tell | Yes | Yes | Yes | Yes | No | Can't tell | No | Minor |
| Muchiri et al., 2024 [48] | Yes | Yes | Yes | Yes | Yes | Can't tell | Yes | No | Minor |
| Van den Burg et al., 2024 [43] | Yes | Yes | Yes | Yes | Yes | No | Yes | No | Minor |
| Campbell et al., 2024 [46] | Yes | Yes | Yes | Yes | Yes | Yes | Yes | No | Minor |

finding is found in S7 Appendix Summary of qualitative findings table. A detailed explanation of the GRADE-CERQual assessment for each review finding is found in the S8 Appendix Evidence profiles.

**Review findings**

We developed 13 descriptive themes which we organized further into two analytical categories: (1) what motivates people to maintain lifestyle changes? and (2) what challenges do people face in maintaining lifestyle changes? Using the locus of control theory, we assessed whether factors that people described as motivating or challenging appeared to be perceived because of their own decisions or actions or as beyond their control (Fig 2).

**Category 1**. **What motivates people with type 2 diabetes to maintain lifestyle changes?**

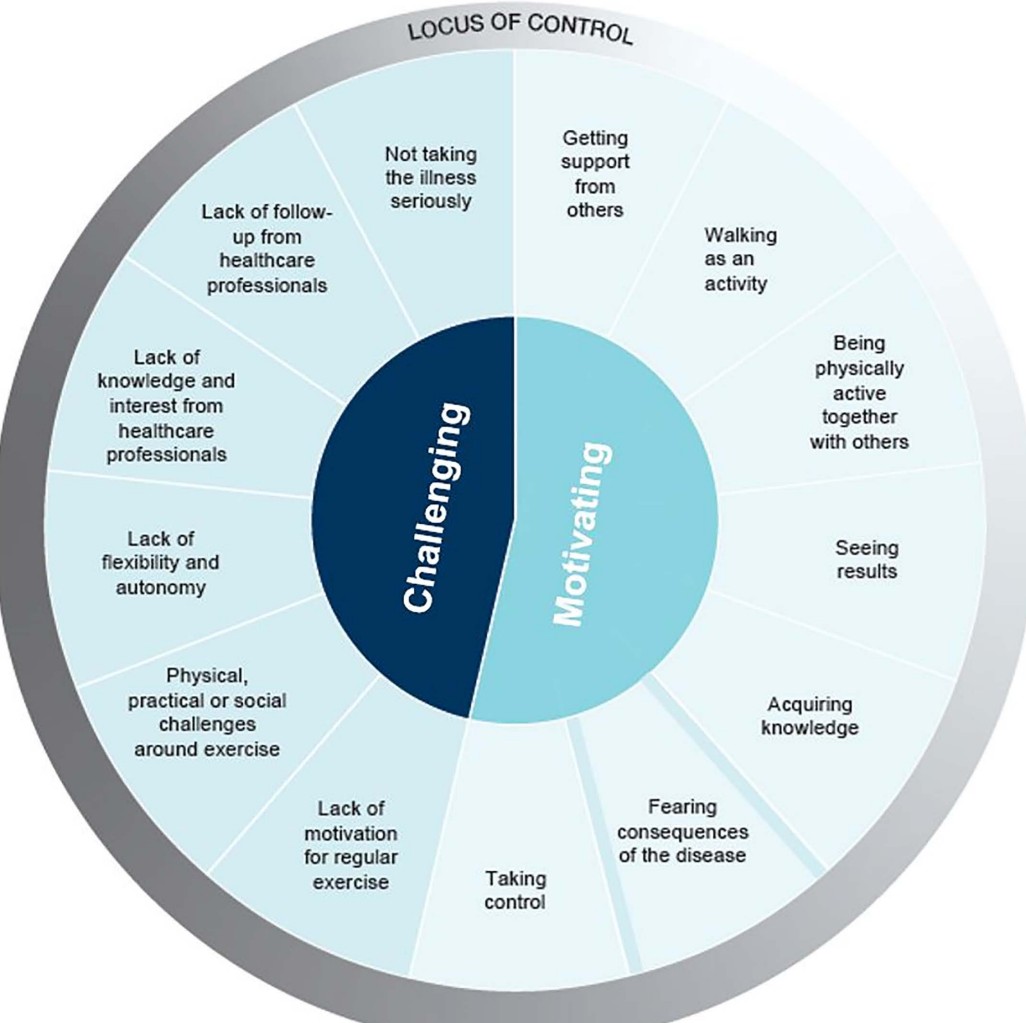

**Fig 2. Motivating and challenging factors when maintaining lifestyle changes according to locus of control.** This figure suggests that factors that people with type 2 diabetes described as motivating to maintain lifestyle changes are also likely to be factors, they believe they have some control over, whereas factors people describe as challenging are likely to be factors they feel they have less control over. This figure also suggest that some factors are a mix of internal and external control, and that one factor (fearing consequences of the disease) is both a motivating and challenging factor.

The participants described several factors that motivated them to maintain lifestyle changes. These included getting support from others, seeing results from their lifestyle changes, walking as an activity and being physically active with others, fearing the consequences of type 2 diabetes and taking control.

**Finding 1. Getting support from others**

**People with type 2 diabetes found it helpful to get support from others when trying to maintain lifestyle changes. "Having someone" around who acted supportively was reported to create commitment and the desire to succeed. This support could come from close relationships, colleagues, exercise groups, healthcare professionals or someone in the same situation. People also perceived tailored support from healthcare professionals as positive for maintaining changes in lifestyle (high confidence, S7 Appendix).**

Several of the participants expressed a need for people in their lives who supported their maintenance of lifestyle changes [36,39,42,45,46]. *"Having someone"* who acted supportive was seen as important for success [37] and this support could come from family [36,38,42], exercise groups or healthcare professionals [38,39,42,44,45,48], spouses, partners [36,39] friends or colleagues [36,46]:

*"People who have diabetes should surround themselves with people who support them... I mean just having someone feel for you the way you are feeling is a great feeling. Having someone in your corner is good" (F4,* [39]).

In Hall et al. [36], colleagues played a significant role in one participant's motivation to continue his lifestyle changes:

*"I use the strategy of disclosing to Ann, my coworker, that I need to test. Ann will ask about my sugar and I don't like to be told or reminded, so this gives me reinforcement to continue."*

Being part of a group of like-minded people who had similar goals, for example with regard to exercise, was experienced as a significant support, as one participant in Conlin et al. [38] described:

*"… So you know to just get mixed up with people who are like-minded and had the same interests in life is important. So, I wouldn't have done it (been successful) by myself. I mean that I had the company of friends; networks; whatever." (M3)*

Support and follow-up from healthcare professionals was mentioned by several participants as important for motivation [38,39,42,46]. Participants saw tailored support that matched their individual needs as most useful [38,42].

**Finding 2. Walking as an activity**

**People with type 2 diabetes found that choosing activities they enjoyed and that fitted into their personal circumstances promoted maintenance of lifestyle changes. Walking as an activity was experienced as simple and with a high degree of feasibility that could be implemented in daily life, for example through exercise, housework, gardening or walking the dog (moderate confidence, S7 Appendix).**

When attempting to be physically active, the participants highlighted the importance of choosing an activity they liked and that fitted into their individual circumstances [36,38,40]. Walking appealed to most of the participants across studies, because the activity was known and considered feasible by the participants [36,39,41]. Going for a walk was an activity that most people had adopted into their daily activities [38–40,43,45,47] and participants mentioned walking as a beneficial part of the lasting lifestyle change [39–41]. Walking was also an activity that was maintained over time and as several participants described, going for a walk had become a positive part of everyday life that was both desirable and achievable [40,41,45]:

*"Still walking 6, 7, 8 miles a day erm every day... It's just become a part of the routine that I enjoy. It's not a chore, it's something that I actually enjoy doing" (Andy,* [41]).

Walking was done in several ways. Some performed walking interval training as part of their daily exercise routine [40], while for others walking was a more integrated part of their daily life, through for example housework and gardening [39].

For some people, having a dog or having access to a dog provided the motivation to achieve regular physical activity and could play a significant role in maintaining their lifestyle change. Walking the dog was also an activity that participants maintained regardless of weather conditions [40]:

*"We get regular exercise every day... Rain, hail or shine... We're out in all weathers aren't we [speaks to the dog]"* (Duncan).

**Finding 3**. **Being physically active with others**

**People with type 2 diabetes saw the company of others, such as joining an exercise group or going for a walk with others, as positive for maintaining physical activity and found it motivating to have regular appointments with or obligations towards others (moderate confidence, S7 Appendix).**

The company of others was mentioned by several participants as a promoting factor for maintaining physical activity [36,38,43,45,48]. Walking with others and exercise groups was mentioned as positive by those who carried out these activities, and as a loss for those who had stopped doing it [40,42]:

*"When it's just me, it's easier to just stay at home, on the sofa. Having commitments and an exercise group motivates me" (Male,* [42]*).*

One participant who had previously been part of an exercise group found it challenging to maintain physical activity after he had dropped out of the group:

*"Nobody is expecting me to do it (physical activity) so quite quickly I'll say, "tomorrow you'll go to the fitness center" and then tomorrow it's postponed until the day after and suddenly a month has gone by" (Participant 3,* [41]*).*

Having obligations towards others was highlighted as a motivating factor and several mentioned regular agreements with other people as positive for maintaining physical activity over time [40,41,43,45].

**Finding 4. Seeing results**

**People with type 2 diabetes described how seeing results from their lifestyle changes had a positive effect on their motivation. This included achievements such as weight loss (high confidence, S7 Appendix).**

Seeing positive results of their lifestyle changes was mentioned by several participants as motivating for maintaining these changes [36,38,41–43,46,48]. Some described a feeling of competence and progress, like this participant in Walker et al. [41]:

*"It is very much the awareness of your own capability of making a difference... You are in charge of what you can do and you have proof that you really can do something" (Participant 3).*

Seeing changes and results related to weight loss was mentioned as particularly motivating [37,41–43,45,46], exemplified by this participant in Walker et al. [41]:

*"I would say a quick weight loss of 5–10 kilos would do wonders. It would really be motivating. You want to get into a loop where you eat right because you are exercising and you are exercising because you are losing weight" (Participant 3).*

Motivation to maintain lifestyle changes was also linked to achieving stable blood sugar levels [36,38,39,45,46,48], improved heart function, reduced pain, less fatigue [35] and increased well-being [36,40,42].

**Finding 5**. **Acquiring knowledge**

**People with type 2 diabetes viewed increased knowledge about the disease as a source of motivation for maintaining their lifestyle changes. Sharing this knowledge with others with the same illness could also strengthen people's commitment to maintain their own lifestyle changes (moderate confidence, S7 Appendix).**

By raising their own level of knowledge about the disease, some participants had succeeded in maintaining their self-care and their lifestyle changes over time [38,39,43,48]. Some described how keeping up-to-date required great effort and was a continuous learning process, as described by this woman [39]:

*"I do a lot of research, I would say every week I am doing at least ten hours of research. I have Google alert for all the new diabetes medical results or scientific inquires. I'm always learning more and keeping up with all new stuff so that's really beneficial" (F3).*

Some participants mentioned that it was important to learn enough about the disease to be able to take control of the situation [39]. Being able to build motivation, positive emotions, and positive action as a result of the acquired knowledge

and experiences, emerged via participants as important for maintaining lifestyle change [38]. Some also mentioned that this knowledge had given them the opportunity to help others in the same situation by being able to tell them about the disease and what it entails, exemplified by this participant [48]:

*"I now can tell others about how to eat when you are a diabetic" (F68).*

This sharing of knowledge also led to an obligation towards themselves to follow disease recommendations over time [36,38].

### Finding 6. Fearing consequences of the disease

**People with type 2 diabetes described how a fear of the complications of diabetes (for instance, a fear of nursing homes, sickness and, death) motivated them to change their lifestyles and maintain these changes over time (moderate confidence, S7 Appendix).**

Participants referred to fear as a motivation for maintaining lifestyle changes [36,39,41,42]. This included fear of being permanently ill, ending up in a nursing home due to their disease [39], and the fear of co-morbidities [41]. Participants across several studies were frightened of the complications [36,39,42] and the fear of dying [36,42] as this participant describes:

*"I would miss a lot of life, I want to live, I want desperately to live, I want quality of life and I don't want to be a burden" (Participant, [36]).*

This fear of complications was a strong driving force that motivated people to change their lifestyles and maintain this change, as described by this woman:

*"I knew that I had to make changes and sustain them. I was well aware of that and was determined to utilize whatever it took to make these changes. The determination came out of fear and the fear was of the complications. I also knew that I had a choice and it is from that choice that I took action" (F2, [39]).*

### Finding 7. Taking control

**People with type 2 diabetes believed that taking control of the disease could help them maintain lifestyle changes. This involved accepting the disease and its consequences, prioritizing oneself, setting goals, and having strategies and structure in everyday life. Some felt that personal qualities such as being competitive, goal-oriented, good at planning, autonomous, mentally strong, self-aware and optimistic were important for taking control and being able to maintain lifestyle change (high confidence, S7 Appendix).**

Several participants pointed out that being successful in maintaining lifestyle changes was about taking control of the disease [38,39,42,47,48]. For some, this involved acknowledging and accepting the disease and its consequences in everyday life [38,42,46]. Taking control was also tied to prioritizing oneself and being dedicated so that the lifestyle change became a habit [39]. Moreover, setting concrete goals was seen as part of taking control and maintaining lifestyle changes [36,41,44,47], for instance avoiding medication use [39]. Others pointed out that the goals *"had to mean something"* [38] and when a chance for change arises, one must seize it:

*"I'd reached a point in my life where I needed a change in my health. This opportunity came along and I took it"(Participant [45]).*

A number of participants pointed out that it was important to have strategies for maintaining control over diet and exercise [36,41]. Diet-related strategies included reducing portions [36,46], eating more salad, drinking water, choosing sugar-free options for desserts or drinks, choosing "low carb" options or avoiding carbohydrates at unplanned events, not having snacks in the house, keeping busy with something, staying away from the kitchen, shortening lunch time, skipping "illegal" food in the shops, or eating before gatherings or events. Participants conveyed that it was important to find out what worked for them over time. Strategies for maintaining physical activity over time included adapting activities to life situations, daily routines, and physical conditions [36].

Having structure in everyday life was also mentioned by several participants as important for retaining control and succeeding in maintaining lifestyle changes [36,39,41,42,48]. Some participants said that they ate at scheduled times, and

that they made pre-planned lists before shopping. Some mentioned that having structure could also mean *"preparing for the unprepared",* such as creating alternative diet plans or adapting exercise plans for different situations. As this participant pointed out, structure in everyday life could ensure that good habits were maintained over time:

*"You can have a lot of intentions about continuing on your own but that can disappear all of a sudden, but then it (structure) will still keep you going".(Participant 4,* [41]).

Several participants also described how certain personal characteristics helped them take control. Being competitive, goal-oriented, good at planning, mentally strong, autonomous, able to lead oneself, and being optimistic were all mentioned as important characteristics [38,39,42].

**Category 2. What challenges the maintenance of lifestyle change among people with type 2 diabetes?**

The participants described several factors that challenged the maintenance of lifestyle changes. Two challenges particularly concerned maintaining regular exercise, including a lack of motivation, or experiencing physical, practical, or social challenges around exercise. Other challenges included the lack of flexibility and autonomy associated with lifestyle changes, a lack of follow-up and a perceived lack of knowledge and interest from healthcare professionals, and not taking the disease seriously.

**Finding 8. Lack of motivation for regular exercise**

**People with type 2 diabetes described several factors that negatively affected their motivation to maintain exercise over time. These included a lack of progress, unachieved goals, and limited understanding of the benefits of physical activity, in addition to a complicated relationship with physical activity from the past, feeling uncomfortable in gyms, the experience of exercise as time-consuming and little enjoyment of the activity (moderate confidence, S7 Appendix).**

To change their lifestyle and maintain this change, some participants said that they had to make major changes in their lives, especially when it came to regular exercise. However, some participants lacked motivation to engage in physical activity [40,42,44,46]. Participants mentioned several factors that affected this motivation negatively. Not seeing progress or not reaching goals was reported to destroy motivation [41] and many cited non-reached weight loss goals as the most challenging [40,41]. Participants also identified the absence of professional support and supervision as an important factor for discontinuation of exercise [45].

Some participants were aware that regular physical activity provided health benefits [40,41], but few participants described exercise as an important part of their lives. Others mentioned a strained relationship with physical activity as the most challenging, or feeling uncomfortable in gyms [40]. A participant in Conlin [38] claimed that he *"treated his car better than he treated his body"* and, like others, stated that exercise was time-consuming [39,41]. The intention and willingness to exercise was also sometimes present in everyday life, without the sessions necessarily being carried out [40].

The commitment to exercise was also dependent on whether participants felt pleasure from doing the actual activity [38,39]. Participant 1 in Walker et al. [40] noted that *"it is important that you are motivated for it [type of physical activity]".* Some participants stated that they lacked joy in exercising, and had to force themselves to continue, like this participant in Peel et al. [40]:

*"I'm still going, but maybe only- like I'll go one week and I'm thinking "ooph"... It's not something I enjoy doing, it's as simple as that. It's like a chore."*

**Finding 9. Physical, practical, or social challenges around exercise**

**People with type 2 diabetes described how their physical condition, in addition to practical and social conditions, made it challenging to exercise regularly (moderate confidence, S7 Appendix).**

Several participants stated that physical limitations or illness made exercise one of the most challenging lifestyle changes to maintain over time [36,39,40]. This could include physical symptoms such as breathing difficulties, fatigue, and pain; chronic illness; or injuries and surgical interventions in connection with injuries or illness [36,40,42]. This participant was one of several with osteoarthritis who talked about their challenges related to exercise:

*"I don't do much exercise because, well mainly because I'm sore with the arthritis" (I2, [40])*.

Some participants described how maintaining changes in diet and exercise took too much time and energy due to practical challenges [42]. These were sometimes related to the participants' work such as changing jobs [36], long working days [39], unfavourable working hours [41] or heavy workloads [42]. One woman said that she lost the opportunity to exercise when her job moved and her commute became 45 minutes longer [40]. Others described practical challenges such as bad weather, fear of snowploughs [36], or storage of exercise equipment as reasons why they saw exercise as a difficult lifestyle change to maintain:

*"I was going to buy a bike, but I don't really have anywhere to keep it" (Callum, [40])*.

Some described how financial or social conditions in their private lives, such as sick parents [37] or bereavement [40], made it challenging to maintain regular exercise. Participants in Hall et al. (2003) [35] mentioned that having a (personal) trainer worked well to maintain physical activity but was too costly. One man in Schmidt et al. [42] described upheavals in private life as challenging:

*"So much has happened, I lost a close friend, my shoulder is injured and I have not been able to exercise the last three months, someone broke into my house, my sister died (….) with all that, sometimes I feel like just giving up" (Male, 74)*.

**Finding 10. Lack of flexibility and autonomy**

**People with type 2 diabetes found that lifestyle changes tied to diet and exercise limited their flexibility. The desire to regain a sense of autonomy and freedom of choice in everyday life created challenges, and it was difficult to see the new lifestyle as "one's own" (low confidence, S7 Appendix).**

Participants stated that maintaining dietary lifestyle changes was difficult because diets or dietary regimes offered little flexibility and came into conflict with the participants' work, daily routines, and personal lives [36,42,45,46], as described by this participant:

*"The hardest part was following the diet programme, all the recommendations and six meals a day. It was hard fitting the meals into my daily routine—that pressure all the time. I couldn't comply; it felt so different all the time" (Male, 72 [42])*.

Maintaining lifestyle changes was also challenged by people's desire to regain freedom of choice and autonomy. Although several participants experienced benefits from changes in exercise and diet, few saw the new lifestyle as "their own"; consequently, lifestyle change was described as challenging to stick to over time [36,42].

**Finding 11. Lack of knowledge and interest from healthcare professionals**

**People with type 2 diabetes experienced that healthcare professionals lacked knowledge about and interest in the disease and its accompanying lifestyle changes (moderate confidence, S7 Appendix).**

Participants experienced a lack of knowledge about the disease and about lifestyle changes among healthcare professionals [39,40] and felt this challenged the collaboration between them [39]. Sometimes the healthcare professionals` lack of knowledge caused confusion among participants [39,40]:

*"They [doctors] say light exercise is good for you, but I don't know if it helps with the diabetes or not"(Walter, [40])*.

Some participants perceived that healthcare professionals lacked interest in type 2 diabetes [40]. They expressed this lack of interest to be a problem for their own maintenance of lifestyle changes [38,39,42], and they felt alone in making lifestyle changes and maintaining them over time [39,40,42]:

*"Well they'll ask, y'know, what exercise you get...but they haven't said "Oh I think you should be walking twice as far" no, nothing like that" (Duncan, [40])*.

**Finding 12. Lack of follow-up from healthcare professionals**

**People with type 2 diabetes described a lack of regular check-ups and follow-up from healthcare professionals. They linked this to a feeling of not being important enough, which affected their maintenance of lifestyle changes (moderate confidence, S7 Appendix).**

Participants described doctors and other health professionals' follow-up of their diabetes type 2 as inadequate [39,42,43], for example by not performing regular check-ups because it was considered this to be "*a waste of time and*

*resources"* [42]. The feeling of not being *"important enough"* was cited as a reason for this lack of follow-up by participants [39]:

*"They see you once then they send you to see a physician's assistant, which is like saying, well you know I wanted to come see you but—I guess it's a self-esteem thinking, you don't feel important enough to be seen. So that's been some of the lows, not finding a good person to work with. Sometimes it's been hard when other people don't seem to understand, but it really doesn't matter what other people think. It's been harder with the medical professionals" (F2, [39]).*

One participant also experienced that she was blamed when the treatment plan did not work properly [42]:

*"I was seeing a lady doctor and I was doing exactly what she told me, which probably brought my stress down. I ate exactly what I was supposed to eat right down to the teaspoonful. I did exactly that, and my sugars still weren't too good. We just couldn't get it down right. Over the phone, and I'm saying I don't know, what can I do? And she'd say, I don't know what you're doing, but you're doing something wrong."*

Participants expressed a desire for more personal engagement and support, highlighting the importance of simple inquiries like `How is it going? Are you taking your walks?` [43,44]. The frequency of consultations could also impact the maintenance of lifestyle changes [46]:

*"Coming off the meal replacements, the meetings with [the dietitian] were not as frequent... You don't have someone there every week making you accountable for your decisions... I probably didn't think about what I was eating as much as I should" (P 01).*

**Finding 13. Not taking the illness seriously**

**People with type 2 diabetes found it challenging to take the disease seriously due to few symptoms. Some described that they didn`t take responsibility for their own health, which made it difficult to maintain lifestyle changes (low confidence, S7 Appendix).**

Although participants pointed out a lack of interest and follow-up by healthcare professionals (confer findings 11 and 12), some participants also described to lack understanding about type 2 diabetes themselves. This was reinforced by experiencing few symptoms of the disease [42]. One participant wished he would feel worse to take the illness seriously:

*"I never had any discomfort from it, and that is a disadvantage, you do not have time to think back upon where you felt really bad (:::) I feel that it motivates me, that my fitness has improved, that walking on stairs is not a problem, that I feel good., I have more energy, I get my sleep, I am fresh when I wake up. Such completely ordinary things, it sounds a little strange, but I actually wish I had felt bad at some point" (Male, 50, [42]).*

Participants also stated that they didn`t take responsibility for their diabetes, which was a barrier to maintaining lifestyle change [38,39]. A participant in Conlin [38] admitted to place responsibility for disease follow-up and lifestyle changes on his doctor, rather than on himself:

*"I consciously took the approach that if he (the personal physician) could tweak the meds it's his problem; not mine. I will tell you that during that entire period as meds kept getting tweaked that I was never involved. I never did anything to make any kind of changes. I just went on down the road" (M2).*

### Motivators, challenges, and locus of control

Using the locus of control theory [10–12,32], we assessed each of the factors that people with type 2 diabetes described as either motivating or challenging for maintaining lifestyle changes, and whether participants appeared to perceive these factors as a result of their own decisions or actions, or as beyond their control. None of the factors has a completely internal or external control location; this is likely to vary based on the individual and the situation. However, our assessment suggests that many of the factors people see as motivating also appear to be factors that they are likely to describe as within their own control (e.g., getting support from others, walking as an activity, seeing results, acquiring knowledge, taking control), while challenging factors are often those that people are likely to describe as outside their control (e.g., experiencing a lack of knowledge and follow up from healthcare professionals). Our assessment suggest that the

motivational factors are things the people actively do themselves, while the challenging factors are things that happen to them. Fearing consequences of the disease seems to be in a class of its own as people do not give themselves fear but do use it actively to motivate themselves. We also suggest that some factors are a mix of internal and external control. We have visualised motivating and challenging factors together with our assessments of locus of control in Fig 2.

## Discussion

### Summary of evidence

Lifestyle interventions have been shown to slow and reverse disease progression in type 2 diabetes. However, sustaining these lifestyle changes over time can be challenging. In this qualitative evidence synthesis, we aimed to explore factors that both contribute to and challenge the maintenance of lifestyle changes in people with type 2 diabetes. Twelve studies met our inclusion criteria.

We identified getting support from others, including family, friends or healthcare professionals, as one factor in sustaining lifestyle changes, such as adhering to dietary and exercise modifications over time. Support from others, especially healthcare professionals, is perceived as motivating and important by people with type 2 diabetes to change their lifestyle [49–51,55] and can also have a positive impact on health outcomes [52]. However, our review findings indicate that people with type 2 diabetes sometimes experienced a lack of support from healthcare professionals; an experience also reported in other studies [49,53,54].

Walking as an activity and being physically active with others were also described by participants as motivating factors. A study evaluating the combination of these factors suggests that people who attend organised walking groups-increase their self-efficacy beliefs, their perceived effectiveness of exercise as a treatment for type 2 diabetes, and to maintain exercise over time [56]. In contrast, challenging factors included lack of motivation due to bad experiences, lack of understanding of the benefits, a feeling that exercise is too time-consuming and a lack of enjoyment for exercise [57]. Thus, people with type 2 diabetes may benefit from participating in low-threshold activities (e.g., walking) with peers as it fosters the social and emotional support necessary to sustain exercise behaviour.

We also identified that fear of the consequences of type 2 diabetes as well as acquiring knowledge about the condition, motivated some people to maintain lifestyle changes. On the other hand, others found it difficult to take the disease seriously. One explanation for the latter finding is offered by Rise et al. [58], who suggest that as the prevalence of type 2 diabetes increases and the diagnosis becomes more common, the negative impacts of poorly managed diabetes may be more difficult to communicate.

Participants in this review described how taking control encouraged them to continue the changes in lifestyle, for instance by setting goals and structuring daily life to overcome obstacles due to their type 2 diabetes. We are not the first to suggest that a person's perceptions of his or her own responsibility is crucial for behavioural change, or that internal locus of control factors stimulate such change. Other studies have found that *"self-activated"* people who acquire knowledge and skills about lifestyle modifications, and whose actions emanate from personal choice and perceived competence, are more successful in adhering to diabetes regimen and improving glycaemic control [11,12,50,51]. There are also some evidence that supports that lifestyle change challenges can be connected to external locus of control and factors outside the persons control, e.g lack of support from both healthcare professionals and non-professionals [18].

### Implications for practice

Below (in Table 3) is a series of questions that build on our findings and that may help people with type 2 diabetes think about how best to maintain their lifestyle choices over time. These questions may also be helpful to healthcare providers when talking to people with type 2 diabetes about making and maintaining lifestyle changes.

**Table 3. Questions that may be helpful when making and maintaining lifestyle changes in type 2 diabetes.**

| Questions that may be helpful when making and maintaining lifestyle changes in type 2 diabetes | |
|---|---|
| **Support from others can be an important part of maintaining lifestyle changes** | 1. Do you have someone (family, friends, colleagues) who can support you in maintaining your lifestyle changes?<br>2. Have you discussed with your doctor or other healthcare professional what kind of support and follow-up you need, and what they can offer? For example, do you need or would you appreciate regular diabetes control? |
| **Physical activity is important for people with type 2 diabetes** | 3. Have you managed to find an activity that you enjoy, in an environment that you feel comfortable in?<br>4. Have you considered finding an activity that you can carry out as part of your daily life? For example, can you use shopping, housework, gardening, or dog walking as a way of getting enough physical activity in your everyday life?<br>5. Have you considered finding an activity that you can do with others? For example, can you join an exercise group or arrange regular walks with someone? |
| **Taking control can help to maintain lifestyle changes** | 6. Have you managed to prioritize yourself and create strategies that can give you structure (for example, eating at scheduled times, planning your shopping lists, making exercise appointments)?<br>7. Do you know what motivates you and have you defined goals that are important to you (disease control, weight loss, blood sugar control or similar)?<br>8. What personal qualities do you think you have that can contribute positively to the maintenance of your maintenance of lifestyle change (for example, are you goal-oriented, good at planning or/and optimistic)? |
| **Knowledge about the disease is important for people with type 2 diabetes and for healthcare professionals who support them.** | 9. Do you and your healthcare provider have ways of keeping up to date with reliable information about diabetes? |

## Implications for future research

The following implications for research are based on our assessment of the studies included in this review and on our GRADE-CERQual assessments of the review findings.

Better reporting is needed in qualitative studies on this topic. Future qualitative studies should transparently report their research methods and should include reflections on researchers' roles and perspectives and how these may have influenced the conduct and results of the study.

Twelve studies are included in this qualitative evidence synthesis. More research about the experiences of people with type 2 diabetes and what motivates them to maintain lifestyle changes over time is needed, especially from low- and middle-income countries.

There is also a need for further research on the following topics and maintenance of lifestyle change; lack of flexibility and autonomy, and not taking the illness seriously.

## Conclusion

To our knowledge, this is the first systematic review that has explored what motivates people with type 2 diabetes to maintain lifestyle changes over time and the challenges they experience. We believe that this review contributes to the therapy area of type 2 diabetes and provides a useful summary of existing evidence. In addition, we have used these findings to develop a series of questions that may help people with type 2 diabetes and their healthcare professionals when managing type 2 diabetes and maintaining lifestyle choices over time.

## Supporting information

**S1 Appendix. ENTREQ Checklist.**
(DOCX)

**S2 Appendix. PRISMA 2020 Checklist.**
(DOCX)

**S3 Appendix. Search strategy.**
(DOCX)

**S4 Appendix. Findings supporting data from studies.**
(XLSX)

**S5 Appendix. Tables of excluded studies.**
(DOCX)

**S6 Appendix. Characteristics and methodological limitations of included studies.**
(DOCX)

**S7 Appendix. Summary of qualitative findings table.**
(DOCX)

**S8 Appendix. Evidence profiles.**
(DOCX)

## Author contributions

**Conceptualization:** Nina Ottesen, Tina Bekken, Lena Victoria Nordheim.

**Data curation:** Nina Ottesen, Tina Bekken.

**Formal analysis:** Nina Ottesen, Tina Bekken, Claire Glenton.

**Investigation:** Nina Ottesen, Tina Bekken.

**Methodology:** Nina Ottesen, Tina Bekken, Claire Glenton, Lena Victoria Nordheim.

**Project administration:** Nina Ottesen, Tina Bekken.

**Supervision:** Claire Glenton, Lena Victoria Nordheim.

**Validation:** Nina Ottesen.

**Visualization:** Nina Ottesen.

**Writing – original draft:** Nina Ottesen, Tina Bekken.

**Writing – review & editing:** Nina Ottesen, Tina Bekken, Claire Glenton, Lena Victoria Nordheim.

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
