## [Decision Letter · Decision Letter 0]

6 Sep 2024

PONE-D-24-00773What motivates people with type 2 diabetes to maintain lifestyle changes and what challenges do they experience? A qualitative evidence synthesis.PLOS ONE?

Dear Dr. Bekken,

Thank you for submitting your manuscript to PLOS ONE. After careful consideration, we feel that it has merit but does not fully meet PLOS ONE’s publication criteria as it currently stands. Therefore, we invite you to submit a revised version of the manuscript that addresses the points raised during the review process.

Thank you for submitting the paper to PLOS ONE. Overall, you have made an effort to identify, appraise, and synthesize qualitative evidence on what motivates people with type 2 diabetes to maintain lifestyle changes. Both reviewers have suggested minor revisions for the manuscript. Please review the comments and make the necessary changes. We look forward to receiving the revised version of the manuscript.

We look forward to receiving your revised manuscript.

Kind regards,

Vishnu Renjith

Academic Editor

PLOS ONE

Journal Requirements:

2. We note that you have referenced (unpublished) on page 6, which has currently not yet been accepted for publication. Please remove this from your References and amend this to state in the body of your manuscript: (ie “Bewick et al. [Unpublished]”) as detailed online in our guide for authors

4. As required by our policy on Data Availability, please ensure your manuscript or supplementary information includes the following: 

Additional Editor Comments:

Thank you for submitting the paper to PLOS ONE. Overall, you have made an effort to identify, appraise, and synthesize qualitative evidence on what motivates people with type 2 diabetes to maintain lifestyle changes. Both reviewers have suggested minor revisions for the manuscript. Please review the comments and make the necessary changes. We look forward to receiving the revised version of the manuscript.

Reviewers' comments:

Reviewer's Responses to Questions

**Comments to the Author**

1. Is the manuscript technically sound, and do the data support the conclusions?

Reviewer #1: Yes

Reviewer #2: Yes

2. Has the statistical analysis been performed appropriately and rigorously?

Reviewer #1: N/A

Reviewer #2: Yes

3. Have the authors made all data underlying the findings in their manuscript fully available?

Reviewer #1: No

Reviewer #2: Yes

4. Is the manuscript presented in an intelligible fashion and written in standard English?

Reviewer #1: Yes

Reviewer #2: Yes

Reviewer #1: I thank the editor for the opportunity given to review this interesting paper.

The authors have taken an effort to review significant papers to conclude it to practical questions needed in the treatment strategy of type -2 diabetes.

Authors claim that this paper is the first of its kind in T2DM management by patients. However, there are such papers already published earlier (doi: 10.1186/1472-6963-14-348).

This paper is a motivated work of the authors to work on an evidence synthesis on T2DM. However, the authors have not included the iterations conducted to reach only six studies, even though there was no restrictions in the range of time limits and languages of the study.

There was no mention about the understanding of the languages of the study published other than English.

There was also no mention about the data management during the process of evidence synthesis.

Recommendation

With this minor revisions, the manuscript can be considered for further process of acceptance for the journal.

Reviewer #2: The area selected for review is good. The reviewers took effort to conduct the detailed review.

It is mentioned articles of all languages selected. Not clear of the language expertise of reviewers.

The six studies included in the final review is from 2003 to 2020, why did they take such a big gap?

The review period is not clear from the article

Different styles of articles compiled together to make the conclusion.

It was mentioned all styles phenomenology, ethnography, grounded theory, case studies, and qualitative process

evaluations were included but those areas are not clear in the review.

**Do you want your identity to be public for this peer review?** For information about this choice, including consent withdrawal, please see our Privacy Policy

Reviewer #1: **Yes: ** Biju Soman

Reviewer #2: No

---

## [Author Response · Author response to Decision Letter 1]

16 Mar 2025

Dear Editor.

Thank you for the opportunity to submit a revised version of our manuscript entitled “What motivates people with type 2 diabetes to maintain lifestyle changes and what challenges do they experience? A qualitative evidence synthesis.” with the manuscript number: PONE-D-24-00773.

We appreciate the feedback on our paper and are grateful for the chance to improve based on the comments received. We also appreciate the deadline extension, which allowed us to extensively update our evidence synthesis and manuscript. This included updating the search, as the previous search was conducted a year ago, leading to the inclusion of additional studies.

The revisions have been integrated into the updated manuscript, addressing all points raised in the Editor's comments. We are hopeful for its consideration and acceptance for publication in PLOS ONE.

Thank you once again for your constructive feedback and guidance.

Sincerely,

Tina Bekken & Nina Ottesen, Claire Glenton and Lena Victoria Nordheim.

---

## [Decision Letter · Decision Letter 1]

29 Aug 2025

What motivates people with type 2 diabetes to maintain lifestyle changes and what challenges do they experience? A qualitative evidence synthesis.

PONE-D-24-00773R1

Dear Dr. Bekken,

We’re pleased to inform you that your manuscript has been judged scientifically suitable for publication and will be formally accepted for publication once it meets all outstanding technical requirements.

Kind regards,

Hidetaka Hamasaki

Academic Editor

PLOS ONE

Additional Editor Comments (optional):

Reviewer #1:

Reviewer #3:

Reviewer #4:

Reviewers' comments:

Reviewer's Responses to Questions

**Comments to the Author**

Reviewer #1: All comments have been addressed

Reviewer #3: (No Response)

Reviewer #4: All comments have been addressed

2. Is the manuscript technically sound, and do the data support the conclusions?

Reviewer #1: Yes

Reviewer #3: Yes

Reviewer #4: Yes

3. Has the statistical analysis been performed appropriately and rigorously?

Reviewer #1: Yes

Reviewer #3: Yes

Reviewer #4: Yes

4. Have the authors made all data underlying the findings in their manuscript fully available?

Reviewer #1: Yes

Reviewer #3: Yes

Reviewer #4: Yes

5. Is the manuscript presented in an intelligible fashion and written in standard English?

Reviewer #1: Yes

Reviewer #3: Yes

Reviewer #4: Yes

Reviewer #1: (No Response)

Reviewer #3: The study entitled " What motivates people with type 2 diabetes to maintain lifestyle changes and what challenges do they experience? A qualitative evidence synthesis." is well written and focuses a very time demanding issue regarding treatment success of a patient with Diabetes Mellitus. The reason behind non compliance on lifestyle modification , in the form of dietary readjustment and regular physical activity shuold be find out, as people knows it very well, but the do not obey it. Knowing is not enough untill it is applied. Twelve article were included in this systematic review, but more could be added and the reason should be more specific including patients inertia.

Reviewer #4: I appreciate your initiative in emphasizing lifestyle change for the prevention and control of Type 2 Diabetes. However, I believe the concept of lifestyle modification should extend beyond food and physical activity alone.

Effective lifestyle change also includes tobacco cessation, limiting or avoiding alcohol use, and stress management. A helpful framework to capture all these elements is SNAPS:

S – Stop Smoking

N – Nutrition

A – Alcohol (Zero Alcohol use)

P – Physical Activity

S – Stress Management

For comprehensive diabetes prevention and control, it is important to consider all aspects of SNAPS. I suggest incorporating this broader perspective into your work so that the message addresses the full spectrum of lifestyle changes needed.

Thank you for your valuable contribution to this important field.

**Do you want your identity to be public for this peer review?** For information about this choice, including consent withdrawal, please see our Privacy Policy

Reviewer #1: No

Reviewer #3: **Yes: ** Farhana Akter

Reviewer #4: **Yes: ** Dr. Momtaz Ahmed,

---

## [Editor Report · Acceptance letter]

PONE-D-24-00773R1

PLOS ONE

Dear Dr. Bekken,

I'm pleased to inform you that your manuscript has been deemed suitable for publication in PLOS ONE. Congratulations! Your manuscript is now being handed over to our production team.

Kind regards,

on behalf of

Dr. Hidetaka Hamasaki

Academic Editor

PLOS ONE